# Oleic Acid-Containing Phosphatidylinositol Is a Blood Biomarker Candidate for SPG28

**DOI:** 10.3390/biomedicines11041092

**Published:** 2023-04-04

**Authors:** Takuya Morikawa, Masatomo Takahashi, Yoshihiro Izumi, Takeshi Bamba, Kosei Moriyama, Gohsuke Hattori, Ryuta Fujioka, Shiroh Miura, Hiroki Shibata

**Affiliations:** 1Division of Genomics, Medical Institute of Bioregulation, Kyushu University, 3-1-1, Maidashi, Higashi-ku, Fukuoka 812-8582, Japan; 2Division of Metabolomics, Medical Institute of Bioregulation, Kyushu University, 3-1-1, Maidashi, Higashi-ku, Fukuoka 812-8582, Japan; 3Department of Nutritional Sciences, Nakamura Gakuen University, 5-7-1, Befu, Jonan-ku, Fukuoka 814-0198, Japan; 4Department of Neurosurgery, Kurume University School of Medicine, 67 Asahi-Machi, Kurume, Fukuoka 830-0011, Japan; 5Department of Food and Nutrition, Beppu University Junior College, 82, Kitaishigaki, Oita 874-8501, Japan; 6Department of Neurology and Geriatric Medicine, Ehime University Graduate School of Medicine, 454, Shitsukawa, Toon 791-0295, Japan

**Keywords:** hereditary spastic paraplegia (HSP), SPG28, *DDHD1*, phospholipase A1 (PLA1), phosphatidylinositol (PI), biomarker

## Abstract

Hereditary spastic paraplegia is a genetic neurological disorder characterized by spasticity of the lower limbs, and spastic paraplegia type 28 is one of its subtypes. Spastic paraplegia type 28 is a hereditary neurogenerative disorder with an autosomal recessive inheritance caused by loss of function of *DDHD1*. *DDHD1* encodes phospholipase A1, which catalyzes phospholipids to lysophospholipids such as phosphatidic acids and phosphatidylinositols to lysophosphatidic acids and lysophoshatidylinositols. Quantitative changes in these phospholipids can be key to the pathogenesis of SPG28, even at subclinical levels. By lipidome analysis using plasma from mice, we globally examined phospholipids to identify molecules showing significant quantitative changes in *Ddhd1* knockout mice. We then examined reproducibility of the quantitative changes in human sera including SPG28 patients. We identified nine kinds of phosphatidylinositols that show significant increases in *Ddhd1* knockout mice. Of these, four kinds of phosphatidylinositols replicated the highest level in the SPG28 patient serum. All four kinds of phosphatidylinositols contained oleic acid. This observation suggests that the amount of oleic acid-containing PI was affected by loss of function of DDHD1. Our results also propose the possibility of using oleic acid-containing PI as a blood biomarker for SPG28.

## 1. Introduction

Hereditary spastic paraplegia (HSP) is a genetic neurological disorder with progressive spasticity of the lower limbs and muscle weakness caused by axonal damage in the pyramidal tract [1,2,3,4,5]. The prevalence of HSP is estimated to be 1.2 to 9.6 per 100,000 [6]. HSP is genetically heterogeneous and is classified into 83 types, at least by 2022, with different responsible genes [7]. HSP is thought to be caused by the obstacles of mitochondrial metabolism, lipid metabolism, membrane transport, axonal transport and myelin formation, and its pathogenesis is extremely diverse [4,6]. Although diagnoses of specific types of HSPs are generally difficult due to the subtle or no differences in their symptoms among types, the molecular mechanisms of pathogenesis can be at least partially different depending on the causative genes. Due to the late onset of HSP [8], it would be useful if we could identify their types with biomarkers before HSP symptoms develop. In this study, we quantitated lipids to identify molecule species showing quantitative changes associated with DDHD1 dysfunction as a potential biomarker for the diagnosis of HSPs.

Spastic paraplegia type 28 (SPG28;OMIM#609340) is an autosomal recessive type of HSP caused by *DDHD1* dysfunction [9,10]. *DDHD1* encodes phospholipase A1 (PLA1), which catalyzes phosphatidic acid (PA) and phosphatidylinositol (PI) to lysophosphatidic acid (LPA) and lysophosphatidylinositol (LPI), respectively. PLA1 specifically hydrolyzes the sn-1 ester bonds of PAs and PIs, producing lysophospholipid with a hydrocarbon group at a sn-2 position, while PLA2, another type of phospholipase A, specifically hydrolyzes the sn-2 ester bonds of PAs and PIs [11,12]. PLA1 can be classified into two types: those that function extracellularly (extracellular PLA1) and those that function intracellularly (intracellular PLA1, iPLA1). DDHD1 is included in the latter type. In mammals, there are three families of iPLA1: DDHD1, p125 and DDHD2 [13,14,15,16]. In humans, DDHD1 is known to be highly expressed in the central nervous system and testes [10,17]. p125 and DDHD2 are expressed in a wide range of tissues outside the nervous system [13,15,16,18]. These three iPLA1s are known to have distinct subcellular localizations [13,16,19]. Since iPLA1 is highly conserved among vertebrates, animal models are useful for observing changes in substrates due to PLA1 deficiency [13].

SPG28 is a very rare genetic disorder that has been reported only from six pedigrees thus far [10,20,21,22]. We have previously reported a novel homozygous 4 bp deletion of the *DDHD1* gene as the responsible variant for SPG28 [22]. We have established *Ddhd1* knockout mice carrying a 5 bp deletion at a very similar position as in the patients and have replicated neurological and histochemical symptoms as seen in the SPG28 patients [23]. Apart from our report, there are two studies that have generated and analyzed *Ddhd1* knockout mice (Figure 1 and Table 1) [17,24]. Both strains lack function of the *Ddhd1* gene by inserting a lacZ trapping cassette and a *neo* cassette into the intron of *Ddhd1,* inducing frameshift [25]. One study reported abnormal sperm motility and mitochondrial morphology [17], while another study reported a significant increase in PI 18:1/20:4 and a significant decrease in LPI 20:4 in the cerebrum of *Ddhd1* knockout mice [24]. Our *Ddhd1* knockout mice also consistently showed significant changes in certain phospholipids in the brain tissues [23,24]. However, quantitative changes of phospholipids have not been examined in other tissues, such as peripheral blood.

HSP is clinically as well as genetically heterogeneous. Neurological examination of patients often fails to distinguish their types without DNA testing [1]. Whole exome sequencing (WES) is commonly used as DNA testing to conduct molecular diagnoses of specific types of HSP [3]. Therefore, if biomarkers specific to particular types of HSP and applicable to patients’ peripheral blood are available, they lead to cheaper and faster molecular diagnosis without DNA testing. Furthermore, biomarkers are potentially useful for examining pathogenic mechanisms, disease progression and drug efficacy.

Biomarkers have been established for some monogenetic diseases, such as phenylketonuria (PKU) and alkaptonuria (AKU). These diseases are caused by defects in genes encoding enzymes *PHA* and *HGD* as in the case of *DDHD1* in SPG28 [26,27]. L-phenylalanine and homogentistic acid are substrates of PHA and HGA, respectively, and have been established as biomarkers for these diseases [28,29]. In monogenic diseases caused by defects in enzyme genes, it is most straightforward to examine the substrates of the enzymes as candidate biomarkers. As quantitative changes of specific phospholipids in *Ddhd1* knockout mice have been reported [23,24], lipid molecules are good candidates for the biomarker of SPG28. Since there are many species of phospholipids with different hydrocarbon groups, it is essential to distinguish molecular species with different side chains by high-resolution lipidome analyses. To address the need for SPG28 biomarkers, here we searched for lipid molecules whose abundances are altered by DDHD1 dysfunction.

## 2. Materials and Methods

### 2.1. Animals

We previously established *Ddhd1* knockout mouse strain (*Ddhd1* (−/−)) by introducing a 5-base-pair deletion in the second exon of the *Ddhd1* gene [23]. The deletion results in a premature termination within the second exon of the *DDHD1* gene, resulting in the expression of a protein completely lacking the DDHD domain. They were fed a standard pellet diet (CLEA Japan, Inc., Tokyo, Japan) and filtered water. The animals were kept under condition of a 12:12 h light:dark cycle and housed in groups of two to five animals per cage. The cages were changed once per week. Mouse husbandry and all mouse experiments were carried out at The Animal Center at Nakamura Gakuen University. All mice were bred in SPF (specific pathogen free) area. Although we did not examine sperm motility abnormalities in our *Ddhd1* (−/−), we maintained the strain by mating *Ddhd1* (+/−) females and *Ddhd1* (+/−) males because they were considered to be infertile based on a previous report [17]. The six individuals in each genotype were chosen completely at random and used for blood collection. Animals used in the experiments were observed by visual inspection to be in normal health status. The background of all mice used in the experiments was C57BL/6J. The animal experiments were conducted with the approval of the Animal Ethics Committee of Nakamura Gakuen University (protocol code: 2016-1; date of approval: 4 April 2016). The study is reported in accordance with the recommendation of the Animal Research: Reporting of In Vivo Experiments (ARRIVE) guidelines.

### 2.2. Genotyping

Tails were cut with lengths of 2 mm at 2 weeks of age for genotyping. DNA was extracted by heating the tail tissues in 50 mM sodium hydroxide solution at 95 °C for 10 min. After heating, 1M Tris-HCl (pH 8.0) was added to neutralize, and the supernatant obtained by centrifugation at 16,000× *g* for 10 min was used as a template for PCR. The second exon of *Ddhd1* was amplified using primers 5′-AAGGTACATCTGGCTTGAAG-3′ and 5′-GAGCTGTGGGTATAGTTGTG-3′. Genotypes were determined by Sanger sequencing of PCR products.

### 2.3. Sample Preparation of Mouse Plasma

Blood samples were collected from the heart of 6-month-old *Ddhd1* (+/+), *Ddhd1* (+/−) and *Ddhd1* (−/−) (*n* = 6 each). Blood collection was performed at 6 months of age in all mice. Both male and female mice were used in a 1:1 ratio to experimental groups. The mice were anesthetized by injecting 10 mL per kg of body weight of mixed anesthesia. Mixed anesthesia was composed by adjusting medetomidine hydrochloride (0.3 mg/kg) (Kyoritsu Seiyaku, Tokyo, Japan), midazolam (4 mg/kg) (Teva Takeda Pharma Ltd., Nagoya, Japan) and butorphanol tartrate (5 mg/kg) (Meiji Seika Pharma Co, Ltd., Tokyo, Japan) with saline (Otsuka Pharmaceutical Co., Ltd., Tokyo, Japan). After complete disappearance of the reflection, the chest was opened, and approximately 1 mL of blood was collected from the left ventricle. The syringe used for blood collection was moistened beforehand with sodium heparin (Mochida Pharmaceutical Co., Ltd., Tokyo, Japan). The blood was collected in a tube containing 1 µL of sodium heparin and centrifuged at 3000× *g* for 20 min at 4 °C. The supernatant was transferred to a new tube and stored as plasma at −80 °C until use.

### 2.4. Human Sample Preparation

Peripheral blood was collected from the SPG28 patient carrying the homozygous 4 bp deletion in the *DDHD1* gene, originally described in Miura et al., 2016 [22], and from unrelated five control individuals free of HSP. The SPG28 patient was a male aged 61 at the time of blood collection. All control individuals were males aged 30, 47, 60, 75 and 76 at the time of the blood collection. Serum was isolated by centrifugation at 3000× *g* for 10 min after blood collection. Peripheral blood was taken following informed consent. The study was approved by the ethics committees of Kyushu University (protocol code: 2020-444; data of approval: 9 October 2020), Faculty of Medicine and Kurume University School of Medicine (protocol code: 21057; the data of approval: 16 June 2021). All methods were carried out in accordance with relevant guidelines and regulations, including the Declaration of Helsinki.

### 2.5. Lipid Extraction

We performed lipid extraction from mouse plasma and human sera using an acidic methanol extraction, as described previously, with minor modifications [23]. Briefly, each 50 μL sample of mouse plasma and human sera was mixed with 1 mL of 20 mM acetic acid in methanol containing the internal standards: PC 15:0/18:1 (d7), 2126 pmol; PA 15:0/18:1 (d7), 102 pmol; PI 15:0/18:1 (d7), 118 pmol; LPC 18:1 (d7), 473 pmol; LPI 17:1, 1714 pmol; and LPA 17:0, 571 pmol. The samples were vortexed for 1 min, and then, sonication was performed for 5 min at room temperature. The samples were centrifuged at 16,000× *g* at 4 °C for 5 min, and 700 μL of the supernatant was transferred in clean tubes. The supernatants were evaporated using a nitrogen gas, and the dried extracts were stored at −80 °C until lipidome analysis.

### 2.6. Lipidome Analysis for Mouse Plasma and Human Sera

The levels of PC, PI, LPC, LPI and LPA in mouse plasma and human sera were quantified using a supercritical fluid supercritical fluid chromatography (SFC) system (Shimadzu Co., Kyoto, Japan) coupled to a triple–quadrupole mass spectrometer (QqQMS) equipped with an electrospray ionization ion source (Shimadzu Co., Kyoto, Japan) as described previously [30]. The SFC system was equipped with a binary pump (i.e., CO_2_ pump and a pump for the modifier and make-up solvent), autosampler, temperature-controlled column oven, and back pressure regulator (BPR). The SFC/QqQMS systems and data acquisition were controlled using Shimadzu LabSolutions ver. 5.99 SP2 software.

All samples were resuspended in 200 μL methanol/chloroform (1:1, *v*/*v*), and 2 μL was injected onto an ACQUITY UPC2™ Torus diethylamine (DEA) column (3.0 mm i.d. × 100 mm, 1.7 μm particle size, Waters Co., Milford, MA, USA). The temperature of the column oven and autosampler was set at 50 and 4 °C, respectively. BPR was set at 10 MPa. The mobile phase was composed of supercritical carbon dioxide (A) and methanol/water (95/5, *v*/*v*) with 0.1% (*w*/*v*) ammonium acetate (B). The mobile phase B was also used for modifier and make-up solvent. The flow rate of the mobile phase and make-up pump was set at 1.0 and 0.1 mL/min, respectively. The chromatographic separation was performed using a gradient of increasing the mobile phase B concentration as follows: 0–1 min: gradient was held at 1% B; 1–24 min; linear gradient of 1% to 75% B; 24–26 min; gradient was held at 75% B; 26–26.1 min; gradient was returned to 1% B; 26.1–30 min; the initial conditions were restored and the column was allowed to equilibrate for 3.9 min.

SFC/QqQMS analyses under multiple-reaction monitoring (MRM) were carried out in the positive-ion mode and in the negative-ion mode (polarity switching mode). The QqQMS parameters were set as follows: electrospray voltage of 4.0 kV in the positive-ion mode and −3.5 kV in the negative-ion mode; heating gas flow rate, 10 L/min; drying gas flow rate, 10 L/min; nebulizing gas flow rate, 3 L/min; heat block temperature, 400 °C; desolvation temperature, 250 °C; detector voltage, 2.3 kV; dwell time, 2 ms; pause time, 2 ms, and polarity switching time of 15 ms. The information of optimized MRM parameters for the lipid molecules is shown in Table 2, Table 3, Table 4 and Table 5 and Appendix A. Identification and quantification of the lipid molecules were performed using Multi-ChromatoAnalysT (Beforce Co., Fukuoka, Japan). The quantitative content of identified lipid molecules was calculated using the ratio of the peak area of each analyte to that of the internal standard of its representative lipid class.

When the objective to examine which molecules among several molecular species changed significantly between *Ddhd1* (+/+) and *Ddhd1* (−/−), the analysis was performed by Student’s t test followed by correction using Benjamini–Hochberg method.

## 3. Results

### 3.1. PC, PA and PI in Mouse Plasma

Phosphatidylcholine (PC), PA and PI in mouse plasma from *Ddhd1* (+/+) and *Ddhd1* (−/−) were analyzed by lipidome analysis. PI and PA are substrates of DDHD1. PC is a precursor of PA, of which the conversion is catalyzed by phospholipase D [31,32]. PCs, PAs and PIs are expected to accumulate due to the loss of DDHD1 enzymatic activity in *Ddhd1* (−/−) mice. A total of 88 different species of PCs and 64 different species of PIs were identified in mouse plasma (Appendix A). PA was not detected in any of the molecular species. Although there was no significant change in the total amount of PC in *Ddhd1* (−/−) (Figure 2A), a significant increase was observed in the total amount of PIs in *Ddhd1* (−/−), as we expected (*p* = 7.2 × 10^−3^) (Figure 2B). We also found significant changes in nine species of PIs, 18:1/20:4, 20:1/20:4, 18:1/20:3, 15:0/24:0, 16:0/18:1, 18:1/18:1, 19:0/20:4, 16:0/20:1 and 18:1/22:5 in *Ddhd1* (−/−) (Table 2 and Appendix A). These quantitative changes were consistently in the same direction of increase in *Ddhd1* (−/−).

### 3.2. LPC, LPA and LPI in Mouse Plasma

We also quantitated PLA1 metabolites, lysophosphatidylcholine (LPC), LPA and LPI in mouse plasma. We detected 72, 8 and 18 species of LPCs, LPAs and LPIs, respectively (Appendix A). Although we observed no significant changes in LPC and LPI in *Ddhd1* (−/−), total LPA was significantly elevated in *Ddhd1* (−/−) (p = 1.1 × 10^−3^) (Figure 3). Among the eight species of LPAs detected in mouse plasma, only LPA 24:0 was significantly elevated in *Ddhd1* (−/−) (p = 0.0004) (Table 3). Although we have previously reported a significant decrease in LPI 20:4 (sn-2) in *Ddhd1* (−/−) cerebra [16], this significant change was not replicated in *Ddhd1* (−/−) plasma from mice (Table 3 and Appendix A).

### 3.3. Focused lipidome Analysis of Human Sera

We performed lipidome analysis on human sera by focusing on nine molecule species significantly changed in the mouse lipidome analyses. Since simple statistical analyses are not applicable due to the unavailability of biological replication of SPG28 patients, we ranked the levels of the seven lipid molecules in the six human samples, including the SPG28 patient. The total amount of PIs observed in the SPG28 patient was within the distribution of the ones in the controls (Figure 4A). Of the nine PIs that showed significant change in *Ddhd1* (−/−) mice, seven PIs, 16:0/18:1, 16:0/20:1, 18:1/18:1, 18:1/20:3, 18:1/20:4, 18:1/22:5 and 20:1/20:4, were also detected in human sera. Among these, four PIs, 18:1/18:1, 18:1/20:3 18:1/24:0 and 18:1/22:5, showed the highest amount in the SPG28 patient (Figure 4B). The amounts of all four PIs in SPG28 consistently exceeded the 95% CI calculated from the control samples (Table 4 and Appendix A). This indicates that these four PIs were elevated in the SPG28 sera. Notably, all four of these PIs contain oleic acid (18:1) in their side chains. LPA 24:0 was not detected in human sera. The SPG28 serum showed the lowest value for total LPAs, in contrast to the elevation of LPAs observed in mice (Figure 4C and Appendix A).

### 3.4. PIs and LPIs Containing Oleic Acid

Since the four PIs showing the highest values in SPG28 commonly contain oleic acid, we examined the amounts of PIs containing oleic acid in mouse plasma and human sera. Oleic acid-containing PIs were significantly elevated in *Ddhd1* (−/−) mice (*p* = 7.0 × 10^−4^) and showed the highest values in the SPG28 patient (Figure 5). We examined the ranks of all oleic acid-containing PIs from the results of the lipidome analysis for human sera. Twelve oleic acid-containing PIs were detected in human sera. Nine of them showed the highest values in SPG28. Furthermore, the levels of these nine oleic acid-containing PIs exceeded the 95% CI of the control samples (Table 5 and Appendix A). This indicates the tendency of an increase in oleic acid-containing PIs in SPG28 sera.

## 4. Discussion

We observed a significant increase in the total amount of PI in *Ddhd1* (−/−) mice that is consistent with the function of PLA1 converting PI into LPI. The increase in total PI is considered to be due to the accumulation of the substrate PI by the failure of the metabolism of PI to LPI due to the loss of DDHD1 function, which is consistent with the prediction based on the metabolic pathway of DDHD1. PI is one of the essential phospholipid components of the plasma membrane. PI is biased toward the inner side of the plasma membrane [33]. Since DDHD1 is a type of intracellular PLA1 (iPLA), the intracellular amount of PI can be greatly affected by the function of DDHD1 [13]. We observed consistent trends of increase in four PIs, 16:0/18:1, 18:1/18:1, 18:1/20:3 and PI 18:1/22:5, both in *Ddhd1* (−/−) plasma from mice and in the human SPG28 serum [23]. Although differences in the content of several phospholipids have been reported between plasma and sera, we assumed the correspondence of quantitative changes in phospholipids between plasma and sera [34]. Notably, all of the PIs that have the highest levels in SPG28 contain oleic acid (18:1) in the side chain, an unsaturated fatty acid with 18 carbons and one double bond. Significant increases in PI 18:1/20:4 have also been observed in the brains of a different strain of *Ddhd1* knockout mice previously published [24]. There was no significant increase in oleic acid-containing PI in *Ddhd1* (+/−) mice, suggesting that sufficient enzymatic activity of DDHD1 remained (Figure 4A). This is consistent with the mode of inheritance of SPG28 being autosomal recessive. Since the increased levels of oleic acid-containing PIs in the peripheral blood are reasonably attributable to the deficiency of DDHD1, our results indicate the possibility of oleic acid-containing PIs as a blood biomarker for SPG28.

LPA is a phospholipid that is abundant in plasma. LPA is also known to be a potential biomarker for ovarian cancer since LPA in plasma is shown to be specifically increased in ovarian cancer patients [35,36,37]. Although LPA is the product of a reaction catalyzed by DDHD1, the total amount of LPA was increased in *Ddhd1* (−/−) mice. Among the eight species of LPAs identified, only LPA 24:0 showed a significant increase, suggesting that the increase in total LPAs is mainly attributable to LPA 24:0. LPA is yielded not only by the PLA1-mediated pathway but also by other pathways involving other enzymes such as lysophospholipase D (LysoPLD) catalyzing lysophoshatidylcholine, lysophosphatidyllethanolamine (LPE) or lysophosphatidylserine (LPS) into LPA [38]. LysoPLD is a secreted enzyme that is a major enzyme-producing PLA in plasma [39]. The excess of LPAs observed in our current analyses is, therefore, attributable to the compensatory pathway such as the one mediated by LysoPLD. Increased activity of LysoPLD and accumulation of LPA are known to occur in female-specific cancers, hepatitis C virus and inflammatory diseases [40,41]. The activity change in LysoPLD is thought to be involved in these diseases in multiple aspects. Increased activity of LysoPLD has been shown to be triggered primarily by inflammatory signals [42]. We therefore speculate that the loss of motoneurons in the pyramidal tract in *Ddhd1* knockout mice can act as inflammatory signals, triggering an increase in LysoPLD activity [23]. Since no significant decrease in LPC 24:0 was observed in the *Ddhd1* (−/−) mice (Appendix A), the observed increase in LPA 24:0 may be supplied by the metabolism of LPE or LPS, which are out of the scope of our focused lipidome analyses. LPA was at the lowest level in the SPG28 patient in contrast to the observation in mice. This is partly attributable to the differences in biological age between the human and mice at examination. The age of 6 months in the *Ddhd1* (−/−) mice is prior to the onset of HSP-like phenotypes and is likely in the middle of the inflammatory process, which may trigger LysoPLD activation without neurological phenotypes. In contrast, since the human sera were collected from the SPG28 patient at the age of 61 after the progression of HSP symptoms, it is possible that the motoneuron degradation prior to the HSP symptoms had reached a plateau and the inflammatory signals might have been weakened.

The significant reduction in LPI 20:4 (sn-2) found in *Ddhd1* (−/−) cerebra was not observed in *Ddhd1* (−/−) plasma from mice (Appendix A) [23]. This discrepancy can be partly attributed to the difference in tissues used for lipid extraction. Plasma is cell-free. DDHD1 is one of the iPLAs and is mainly involved in the intracellular metabolism of phospholipids [43]. Therefore, it is likely that the lipidome analysis using plasma, which is cell-free, does not directly reflect the effects of intracellular DDHD1 metabolism. Since DDHD1 seems to be functionally defective in blood cells, a decrease in LPI 20:4 (sn-2) as in the cerebra is likely to occur.

SPG39, SPG54 and SPG56 are also caused by defects in enzymes involved in PI metabolism, *PNPLA6*, *DDHD2* and *CYP2U1*, respectively [10]. Notably, *DDHD2* is responsible for SPG54 functions such as in iPLA1 as well as in *DDHD1* [44,45]. Since abnormal PI metabolism is likely to be the common pathological mechanism for these types of HSPs, peripheral PI levels are good candidates for biomarkers of other HSPs sharing pathogenesis with SPG28.

The major problem with the current study is that statistical analyses were not applicable to human samples due to the unavailability of additional SPG28 samples, since the disease is extremely rare. Although further studies of specificity and sensitivity of oleic acid-containing PIs using additional SPG28 patients are required for establishment as a biomarker, oleic acid-containing PIs may still be useful to group SPGs sharing similar pathogenic pathways and may still help in clarifying their pathogenic mechanisms, potentially contributing to the treatment. Identification of the blood biomarker is useful not only for diagnosis of HSP types in clinical practice but also as a platform for therapeutic drug development. For example, we should be able to evaluate the effect of a drug by measuring oleic acid-containing PI after administering an SPG28 therapeutic agent to patients. To achieve these goals, it is essential to establish standard values for the amount of oleic acid-containing PI. Measurements of oleic acid-containing PI in larger numbers of SPG28 patients and establishment of the standard values are important tasks for this goal.

## 5. Conclusions

Lipidome analysis of SPG28 model mice and a SPG28 patient was performed to search for a biomarker in blood that is useful for the diagnosis of SPG28. Lipidome analysis of mouse plasma revealed a significant increase in total PI and LPA in *Ddhd1* (−/−) mice. We examined the changes in each PI and identified nine PIs that were significantly elevated in *Ddhd1* (−/−) mice. Next, we examined the level of these nine PIs and total PA in human serum. Seven of the nine PIs found to be elevated in *Ddhd1* (−/−) mice were also detected in human sera. Four of these has the highest level in the SPG28 patient, and all of them were oleic acid-containing PIs (PI 18:1/18:1, PI 18:1/20:3, PI 18:1/20:4 and PI 18:1/22:5). LPA showed lower levels in the SPG28 patient, different from the results observed in *Ddhd1* (−/−) mice. From our observation, we propose that oleic acid-containing PI is a promising blood biomarker for SPG28.

## Figures and Tables

**Figure 1 biomedicines-11-01092-f001:**
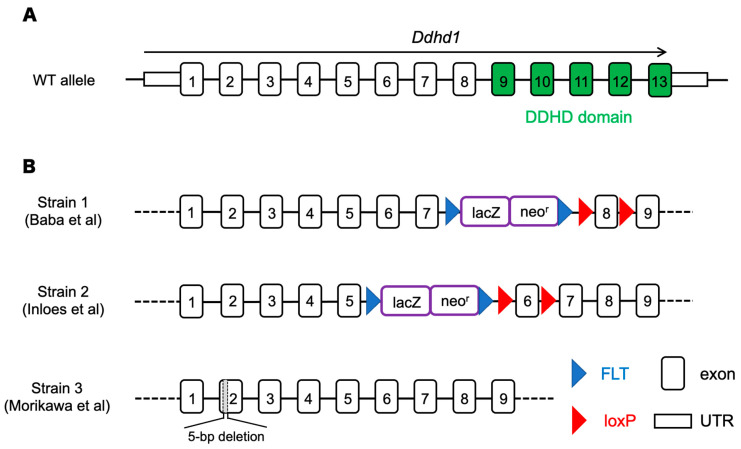
(**A**) The structure of *Ddhd1* gene (NM_176845.5). Exons including the region of the DDHD domain that is critical for enzymatic activity are shown in green. (**B**) *Ddhd1* knockout mice established previously. Strain 1 and 2 deleted the *Ddhd1* gene by inserting a cassette of lacZ and neomycin resistance gene inside an intron [17,24]. In strain 3, *Ddhd1* is disrupted by inducing 5 bp deletion to induce premature termination [23]. The map is not drawn to scale.

**Figure 2 biomedicines-11-01092-f002:**
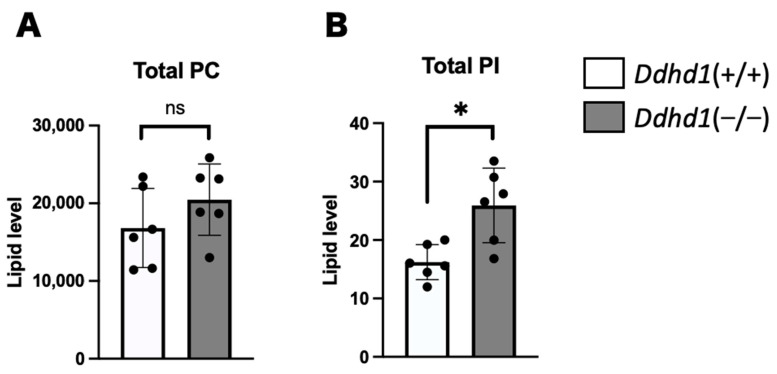
Comparison of total amount of PC and PI. Total amount of PC (**A**) and PI (**B**) in plasma from mice (n = 6 each). *Ddhd1* (+/+) mice and *Ddhd1* (−/−) mice are shown in opened and filled columns, respectively. The unit of the vertical axis is pmol/mg. Error bars indicate mean ± SD. ns: not significant, *: *p* < 0.05. All data were analyzed by Student’s *t* test.

**Figure 3 biomedicines-11-01092-f003:**
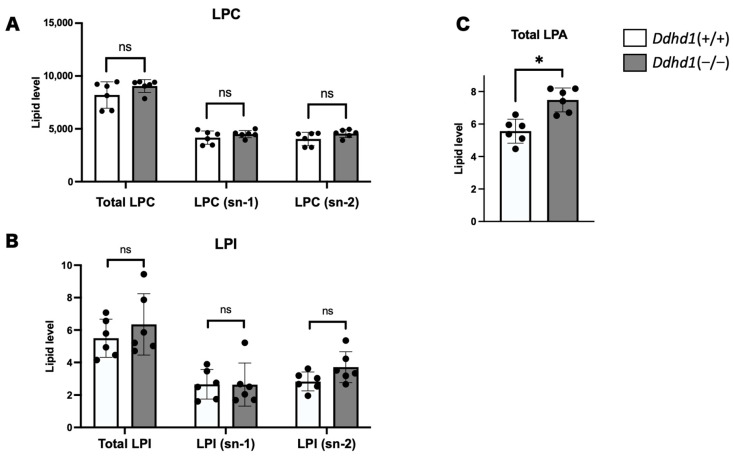
Comparison of total amount of LPC, LPI and LPA. Amount of LPC (**A**), LPI (**B**) and LPA (**C**) in plasma from mice. LPA is shown in only total amount due to the inseparability of sn-1 and sn-2 in our analyses. *Ddhd1* (+/+) mice and *Ddhd1* (−/−) mice are shown in opened and filled columns, respectively. The unit of the vertical axis is pmol/mg. Error bars indicate mean ± SD. ns: not significant, *: *p* < 0.05. All data were analyzed by Student’s *t* test.

**Figure 4 biomedicines-11-01092-f004:**
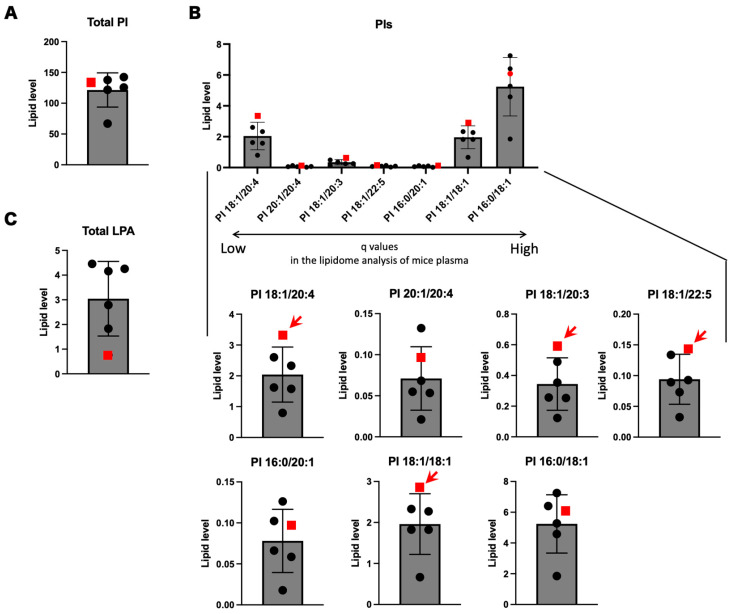
Lipidome analysis using human sera of the SPG28 patient and five controls. (**A**) Total PIs in human sera. (**B**) Focused analyses of human sera on PI species significantly changed in *Ddhd1* (−/−) plasma from mice. PIs arranged in the order of decreasing q values obtained from lipidome analysis using plasma from mice. Individually enlarged charts are shown below. Red arrows indicate the PI species with the highest levels in the SPG28 patient. (**C**) Total LPAs in human sera. The data points of SPG28 are represented by red squares. The unit of the vertical axis is pmol/mg.

**Figure 5 biomedicines-11-01092-f005:**
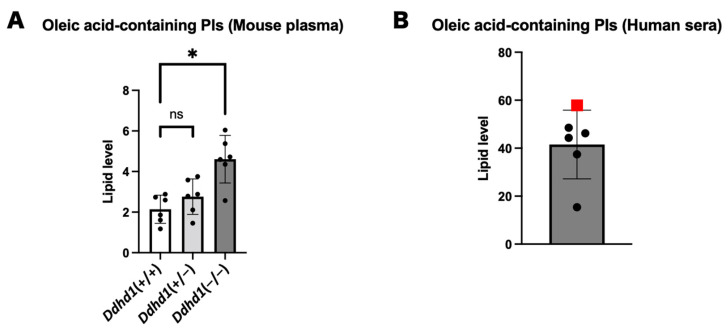
PIs and LPAs containing oleic acid. (**A**) Total oleic acid-containing PIs in plasma from mice. ns: not significant. The data were analyzed by Dunnett’s multiple comparison test. (**B**) Total oleic acid-containing PIs in human sera. The data point of SPG28 is represented by a red square. The unit of the vertical axis is pmol/mg. Error bars indicate mean ± SD. *: *p* < 0.05.

**Table 1 biomedicines-11-01092-t001:** Observed phenotypes of three different *Ddhd1* knockout mice [17,23,24].

Strain	Phenotypes
Strain 1 (Baba, 2014 [17])	・Shortened mitochondrial sheath in sperm ・Impaired sperm motility
Strain 2 (Inloes, 2018 [24])	・Accumulation of PI 18:1/20:4 ・Decrease of LPI 20:4
Strain 3 (Morikawa, 2021 [23])	・Gait disturbance-like symptpms ・Decrease of LPI 20:4 (sn-2) in cerebra ・Axonal decrease in the pyramidal tract

**Table 2 biomedicines-11-01092-t002:** PIs identified from mice plasma.

PIs	Retention Time (min)	*Ddhd1* (+/+)		*Ddhd1* (−/−)		*p*-Value	*q*-Value	
		Average	±SD	Average	±SD			
PI 18:1/20:4	17.19 ± 0	0.645	0.131	1.495	0.271	0.0001	0.006	*
PI 20:1/20:4	17.08 ± 0.01	0.0141	0.005	0.0410	0.011	0.0005	0.017	*
PI 18:1/20:3	17.13 ± 0	0.153	0.043	0.435	0.129	0.0009	0.020	*
PI 18:1/22:5	17.2 ± 0.01	0.0258	0.013	0.0782	0.027	0.0029	0.046	*
PI 19:1/20:4	17.13 ± 0.01	0.00777	0.003	0.0239	0.009	0.0034	0.046	*
PI 16:0/20:1	16.94 ± 0.01	0.0141	0.006	0.0367	0.012	0.0041	0.046	*
PI 15:0/24:0	17.28 ± 0	0.0524	0.030	0.1668	0.062	0.0041	0.046	*
PI 18:1/18:1	17.02 ± 0	0.185	0.078	0.370	0.085	0.0052	0.046	*
PI 16:0/18:1	17.03 ± 0	0.214	0.065	0.437	0.126	0.0057	0.046	*
PI 18:1/22:6	17.28 ± 0	0.0846	0.049	0.238	0.092	0.0079	0.051	
PI 16:0/18:2	17.15 ± 0	0.829	0.219	1.61	0.493	0.0088	0.051	
PI 18:2/20:1	17.03 ± 0.01	0.0131	0.006	0.0284	0.009	0.010	0.053	
PI 16:0/18:0	17.01 ± 0.02	0.0196	0.006	0.0375	0.011	0.011	0.055	
PI 17:0/20:3	17.09 ± 0.02	0.0205	0.006	0.0343	0.008	0.012	0.055	
PI 18:1/20:5	17.3 ± 0	0.201	0.094	0.397	0.111	0.013	0.055	
PI 18:1/18:2	17.14 ± 0	0.376	0.150	0.751	0.239	0.014	0.056	
PI 16:0/16:0	17.08 ± 0.01	0.0194	0.007	0.0296	0.004	0.016	0.061	
PI 16:0/20:4	17.2 ± 0	1.20	0.233	2.25	0.788	0.017	0.061	
PI 20:1/22:6	16.23 ± 3.93	0.00158	0.001	0.00571	0.003	0.021	0.069	
PI 16:0/22:6	17.28 ± 0	0.308	0.164	0.749	0.325	0.022	0.070	
PI 15:0/22:5	17.24 ± 0	0.0243	0.007	0.0471	0.017	0.022	0.070	
PI 18:1/18:3	17.23 ± 0.01	0.00816	0.006	0.0187	0.007	0.023	0.070	
PI 16:0/20:3	17.13 ± 0	0.318	0.098	0.655	0.269	0.025	0.070	
PI 17:0/18:2	17.1 ± 0.01	0.0327	0.010	0.0580	0.020	0.029	0.079	
PI 16:1/18:2	17.26 ± 0.01	0.0228	0.018	0.0557	0.023	0.032	0.081	
PI 16:0/22:5	17.2 ± 0	0.0888	0.048	0.227	0.115	0.032	0.081	
PI 16:1/20:4	17.31 ± 0.01	0.0105	0.006	0.0338	0.020	0.035	0.082	
PI 18:0/18:0	16.92 ± 0.01	0.0272	0.007	0.0419	0.012	0.037	0.084	
PI 18:0/20:4	17.07 ± 0	5.60	0.756	7.11	1.174	0.037	0.084	
PI 18:0/20:2	16.99 ± 0.01	0.0975	0.028	0.1571	0.048	0.037	0.084	
PI 18:0/18:1	16.93 ± 0	0.226	0.061	0.337	0.084	0.037	0.084	
PI 18:0/20:3	17.01 ± 0.01	1.32	0.350	1.91	0.434	0.041	0.084	
PI 15:0/22:2	16.98 ± 0	0.0265	0.010	0.0469	0.017	0.042	0.084	
PI 18:2/23:0	17.16 ± 0.02	0.00120	0.001	0.00268	0.001	0.042	0.084	
PI 18:0/20:1	16.84 ± 0.02	0.0156	0.006	0.0279	0.011	0.046	0.084	
PI 18:2/20:4	17.29 ± 0.02	0.0274	0.014	0.0543	0.023	0.048	0.086	
PI 18:0/18:2	17.02 ± 0	1.09	0.265	1.55	0.383	0.050	0.086	
PI 16:0/18:3	17.24 ± 0.01	0.0168	0.007	0.0341	0.017	0.058	0.097	
PI 18:0/18:3	17.12 ± 0.01	0.0167	0.007	0.0265	0.007	0.059	0.097	
PI 18:2/25:0	17.06 ± 0.05	0.00225	0.001	0.00328	0.001	0.059	0.097	
PI 19:0/20:4	17 ± 0	0.0809	0.029	0.131	0.046	0.064	0.10	
PI 16:1/18:1	17.15 ± 0	0.0182	0.012	0.0393	0.019	0.066	0.10	
PI 16:0/20:5	17.31 ± 0	0.389	0.170	0.764	0.370	0.067	0.10	
PI 16:0/16:1	17.16 ± 0.01	0.0198	0.011	0.0389	0.018	0.075	0.11	
PI 22:1/22:6	16.32 ± 3.95	0.000925	0.001	0.00280	0.002	0.078	0.11	
PI 19:0/22:6	17.12 ± 0.02	0.00492	0.003	0.0095	0.005	0.080	0.11	
PI 18:0/22:6	17.16 ± 0	0.712	0.286	1.122	0.377	0.081	0.11	
PI 18:2/19:0	16.97 ± 0	0.0180	0.009	0.0323	0.014	0.090	0.12	
PI 18:2/18:2	17.25 ± 0	0.0799	0.051	0.1649	0.088	0.091	0.12	
PI 15:0/22:0	17.41 ± 0.02	0.0116	0.010	0.0232	0.011	0.11	0.14	
PI 17:0/20:4	17.13 ± 0	0.0969	0.014	0.1305	0.041	0.12	0.14	
PI 17:0/18:1	16.98 ± 0.01	0.0092	0.003	0.0149	0.007	0.12	0.15	
PI 20:0/20:4	16.96 ± 0.01	0.0302	0.023	0.0568	0.027	0.12	0.15	
PI 15:0/22:1	17 ± 0.22	0.00741	0.004	0.0126	0.006	0.13	0.15	
PI 18:0/22:4	16.97 ± 0	0.0325	0.011	0.0478	0.017	0.13	0.15	
PI 19:0/20:3	16.96 ± 0.01	0.0154	0.009	0.0268	0.013	0.13	0.15	
PI 18:0/22:5	17.09 ± 0.01	0.181	0.083	0.278	0.103	0.13	0.15	
PI 15:0/24:1	17.37 ± 0.02	0.0067	0.006	0.0140	0.009	0.16	0.18	
PI 16:1/18:0	17.02 ± 0.01	0.0175	0.007	0.0251	0.008	0.16	0.18	
PI 18:0/20:5	17.18 ± 0	1.10	0.229	1.33	0.324	0.22	0.23	
PI 17:0/20:5	17.25 ± 0	0.0259	0.011	0.0365	0.015	0.23	0.24	
PI 20:4/22:1	17.21 ± 0.01	0.00481	0.002	0.0065	0.003	0.28	0.29	
PI 20:0/22:6	14.22 ± 6.35	0.00153	0.002	0.00233	0.001	0.46	0.47	
PI 20:5/24:1	17.19 ± 0.03	0.00325	0.001	0.00305	0.002	0.84	0.84	

The unit of measurement is pmol/mg. The *q* value was calculated by Student’s t-= test followed by correction by the Benjamini–Hochberg method. * *q* < 0.05.

**Table 3 biomedicines-11-01092-t003:** LPAs identified from mice plasma.

LPAs	Retention Time	*Ddhd1* (+/+)		*Ddhd1* (−/−)		*p*-Value	*q*-Value	
	(min)	Average	SD	Average	SD			
LPA 24:0	22.89 ± 0.04	1.17	0.47	3.12	0.45	0.0001	0.0004	*
LPA 18:0	22.89 ± 0.03	0.329	0.04	0.437	0.07	0.015	0.061	
LPA 22:6	22.89 ± 0.13	1.48	0.18	1.38	0.10	0.29	0.78	
LPA 22:5	22.84 ± 0.06	0.905	0.10	0.879	0.04	0.61	>0.99	
LPA 20:2	22.89 ± 0.02	0.275	0.07	0.267	0.04	0.83	>0.99	
LPA 20:4	22.32 ± 0.06	0.0825	0.02	0.0786	0.03	0.84	>0.99	
LPA 20:3	22.89 ± 0.03	0.778	0.18	0.796	0.11	0.86	>0.99	
LPA 18:2	22.89 ± 0.03	0.546	0.24	0.531	0.10	0.90	>0.99	

The unit of measurement is pmol/mg. The *q* value was calculated by Student’s t test followed by correction by the Benjamini–Hochberg method. * *q* < 0.05.

**Table 4 biomedicines-11-01092-t004:** Results in human sera for PIs significantly changed in *Ddhd1* (−/−).

PIs	Retention Time (min)	SPG28	Control 1	Control 2	Control 3	Control 4	Control 5	Rank of SPG28 among All Samples	Average of Conrols	SD of Conrols	95% CI		SPG28 out of 95% CI?
PI 18:1/20:4 *	17.20 ± 0.01	3.33	2.60	1.58	1.62	2.33	0.797	1	1.79	0.634	0.556	(1.23–2.34)	Yes
PI 20:1/20:4	17.10 ± 0.02	0.0963	0.0534	0.0549	0.0684	0.1323	0.0212	2	0.0660	0.037	0.0321	(0.0340–0.0981)	No
PI 18:1/20:3 *	17.17 ± 0.01	0.589	0.489	0.253	0.257	0.353	0.123	1	0.295	0.121	0.106	(0.189–0.401)	Yes
PI 18:1/22:5 *	17.21 ± 0.01	0.143	0.134	0.0928	0.0730	0.0892	0.0325	1	0.0843	0.0327	0.0287	(0.0556–0.113)	Yes
PI 16:0/20:1	16.97 ± 0.01	0.0971	0.0661	0.0586	0.1023	0.1261	0.0178	3	0.0742	0.0374	0.0328	(0.0414–0.107)	No
PI 18:1/18:1 *	17.04 ± 0.01	2.85	2.27	1.83	2.33	1.82	0.665	1	1.78	0.598	0.524	(1.26–2.31)	Yes
PI 16:0/18:1	17.06 ± 0.01	6.08	6.40	5.28	7.25	4.59	1.85	3	5.07	1.85	1.62	(3.45–6.70)	No

PIs are arranged in describing order of q value obtained from lipidome analysis in mice. The unit of measurement is pmol/mg. * PIs that have the highest level in the SPG28 patient.

**Table 5 biomedicines-11-01092-t005:** Oleic acid-containing PIs in human sera.

PIs	Retention Time (min)	SPG28	Control 1	Control 2	Control 3	Control 4	Control 5	Rank of SPG28 among All Samples	Average of Conrols	±SD of Conrols	95% CI		SPG28 out of 95% CI?
PI 16:1/18:1	17.17 ± 0.01	0.374	0.362	0.306	0.252	0.415	0.0819	2	0.283	0.114	0.100	(0.183–0.384)	No
PI 16:0/18:1	17.06 ± 0.01	6.08	6.40	5.28	7.25	4.59	1.85	3	5.07	1.85	1.62	(3.45–6.70)	No
PI 17:0/18:1	17.00 ± 0.01	0.288	0.316	0.179	0.212	0.211	0.0951	2	0.203	0.0710	0.0622	(0.141–0.265)	No
PI 18:3/18:1	17.28 ± 0.02	0.125	0.0620	0.0826	0.0281	0.0316	0.0172	1	0.0443	0.0243	0.0213	(0.0230–0.0656)	Yes
PI 18:1/18:2	17.17 ± 0.01	3.25	1.93	2.33	1.59	2.19	0.803	1	1.77	0.544	0.477	(1.29–2.24)	Yes
PI 18:1/18:1	17.04 ± 0.01	2.85	2.27	1.83	2.33	1.82	0.665	1	1.78	0.598	0.524	(1.26–2.31)	Yes
PI 18:0/18:1	16.95 ± 0.01	11.2	9.47	10.0	9.28	6.50	3.17	1	7.68	2.57	2.25	(5.44–9.93)	Yes
PI 18:1/20:5	17.32 ± 0.02	0.107	0.0714	0.0916	0.0444	0.0362	0.0200	1	0.0527	0.0256	0.0224	(0.0303–0.0751)	Yes
PI 18:1/20:4	17.20 ± 0.01	3.33	2.60	1.58	1.62	2.33	0.797	1	1.79	0.634	0.556	(1.23–2.34)	Yes
PI 18:1/20:3	17.17 ± 0.01	0.589	0.489	0.253	0.257	0.353	0.123	1	0.295	0.121	0.106	(0.189–0.401)	Yes
PI 18:1/22:6	17.30 ± 0.01	0.250	0.138	0.132	0.161	0.156	0.0502	1	0.127	0.0400	0.0351	(0.0922–0.162)	Yes
PI 18:1/22:5	17.21 ± 0.01	0.143	0.134	0.0928	0.0730	0.0892	0.0325	1	0.0843	0.0327	0.0287	(0.0556–0.113)	Yes
Total Oleic acid-containing PI		28.6	24.3	22.2	23.1	18.7	7.70	1	19.2	6.03	5.29	(13.9–24.5)	Yes
Total PI		134	138	142	122	126	67.0	3	119	27.1	23.7	(95.2–143)	No

## Data Availability

The datasets used and/or analyzed during the current study are available from the corresponding author on reasonable request, while all the analyzed data are included within this article and its Appendix A.

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
