# Peer review of "Oleic Acid-Containing Phosphatidylinositol Is a Blood Biomarker Candidate for SPG28"

_biomedicines, 2023, doi:10.3390/biomedicines11041092_

Round 1

Reviewer 1 Report

Morikawa et al perform lipidome analyses on plasma from SPG28 KO mice and then examine those lipids that show significant changes in the sera from an SPG28 patient. They identify 4 oleic acid containing PIs as potential peripheral biomarkers for this disorder.

The paper is interesting and is a good beginning to identifying biomarkers for SPG28 at least in mice. Although the data seem to be consistent across mice and the one human sample, because it is only an n=1 sample size, the data is suggestive but not conclusive (which the authors state in the Discussion).

Is there any way for the authors to get sera from another patient? Or a different sample from this same patient to see if the findings are consistent? Or is it possible to get plasma from a different mouse model (in the Intro they state there are 3 mouse models) to see of the data are consistent? Or lastly, in the Discussion they mention that a few other SPGs (39,54 and 56) are also caused by defects in PI metabolism—anyway to get sera from any of these patients to see if this is something consistent across SPGs.

There are several mistakes within the body of the paper which may be due to formatting? For instance, in Fig 1 and 2 there is no difference between the bars (need to add fill).

Also in Fig 1, 2 and 4 there is no * denoting significance on top of bars in 1B, 2C and 4A.

In Table 1 and 2 the column headed by Ddhdl (+/-) should be (+/+).

Not sure the ratio order matters but in the text under Results (pg 4), one of the nine species with significant change is called 20:4/20:1 whereas in Table 1 it is PI 20:1/20:4. It would also be easier if the text order listing the 9 species matched the order they are listed in Table 1.

In Results 3.3 the first sentence says they will focus on the 9 species with significant change but then the next sentence says we ranked the seven lipid molecules in the six human samples? In this same section, it lists the four PIs that show highest amount in the patient but comparing this to Figure 3, one of these seems wrong, 16:0/18:1 is listed in the text but Figure 3 shows 18:1/20:4 to be the significant species.

Table 4 lists 10 species as Yes (SPG28 out of 95% CI) instead of 9 as stated in the text (Results 3.4). I think the 3rd row should be No rather than Yes?

Author Response

Morikawa et al perform lipidome analyses on plasma from SPG28 KO mice and then examine those lipids that show significant changes in the sera from an SPG28 patient. They identify 4 oleic acid containing PIs as potential peripheral biomarkers for this disorder.

The paper is interesting and is a good beginning to identifying biomarkers for SPG28 at least in mice. Although the data seem to be consistent across mice and the one human sample, because it is only an n=1 sample size, the data is suggestive but not conclusive (which the authors state in the Discussion).

> Thank you for appreciating our paper! All changes from the first draft are highlighted in yellow. We responded to all comments as follows:

  1. Is there any way for the authors to get sera from another patient? Or a different sample from this same patient to see if the findings are consistent?

> Although it is very important to measure other SPG28 patient samples, only six cases of SPG28 patients have been reported worldwide including the case reported by us in Japan. We also understand the importance of validation of the elevated oleic acid-containing PI seen in this study using additional SPG28 patients, but it is extremely difficult to obtain additional SPG28 specimen. It is also quite difficult to obtain another blood sample from the current patient due to the transfer of the physician in charge.

  1. Or is it possible to get plasma from a different mouse model (in the Intro they state there are 3 mouse models) to see of the data are consistent?

> It would be very interesting to see if the increase in PI in different strains of ddhd1 knockout mice. However, due to the retirement of the collaborator who managed the mouse room, we are currently unable to use the mouse facility for the additional mouse experiments.

  1. Or lastly, in the Discussion they mention that a few other SPGs (39,54 and 56) are also caused by defects in PI metabolism—anyway to get sera from any of these patients to see if this is something consistent across SPGs.

> We checked sera collected from two HSP patients that we have ascertained. Unfortunately, we do not possess samples from SPGs 39, 54, and 56.

  1. There are several mistakes within the body of the paper which may be due to formatting? For instance, in Fig 1 and 2 there is no difference between the bars (need to add fill).

> We apologize for any inconvenience caused. The fill seems to have disappeared when converting to PDF. I have corrected Figures 1 and 2 as noted. (Due to adding new figure, Figure1 and 2 have been changed to Figure 2 and 3 respectively.)

  1. Also in Fig 1, 2 and 4 there is no * denoting significance on top of bars in 1B, 2C and 4A.

> We apologize for any inconvenience caused. The diagram seems to have been changed when converting to pdf. I have corrected Figures 1, 2 and 4 as noted. (Due to adding new figure, Figure1, 2 and 4 have been changed to Figure 2, 3 and 5 respectively.)

  1. In Table 1 and 2 the column headed by Ddhdl (+/-) should be (+/+).

> Thank you for bringing up this important point. I have corrected Ddhd1 (+/-) to Ddhd1 (+/+) in Table1, 2 (Due to adding new Table, Table1 and 2 have been changed to Table 2 and 3.)

  1. Not sure the ratio order matters but in the text under Results (pg 4), one of the nine species with significant change is called 20:4/20:1 whereas in Table 1 it is PI 20:1/20:4. It would also be easier if the text order listing the 9 species matched the order they are listed in Table 1.

> Thank you for bringing up this important point. We unified the notation so that those with fewer carbons and double bonds are shown on the left (L233).

  1. In Results 3.3 the first sentence says they will focus on the 9 species with significant change but then the next sentence says we ranked the seven lipid molecules in the six human samples? In this same section, it lists the four PIs that show highest amount in the patient but comparing this to Figure 3, one of these seems wrong, 16:0/18:1 is listed in the text but Figure 3 shows 18:1/20:4 to be the significant species.

> Thank you for pointing this out. There was a typographical error in the manuscript. The 16:0/18:1 in the text is incorrect; 18:1/20:4 is correct. The relevant part has been corrected (L278-L279).

  1. Table 4 lists 10 species as Yes (SPG28 out of 95% CI) instead of 9 as stated in the text (Results 3.4). I think the 3rdrow should be No rather than Yes?

> Thank you for pointing this out. The relevant part of Table 4 has been corrected. (Due to adding new Table, Table4 has been changed to Table 5.)

Reviewer 2 Report

Estimated Authors,

Thank you for the opportunity to review this interesting and innovative paper.

In this research study, Authors have addressed the option for new and innovative testing strategies based on the assessment of potential biomarkers for the spastic paraplegia type 28. The rarity of this disorder, and the difficulties experienced in obtaining valuable samples is appropriately addressed across the main text, and the very rare occurrence of this disease is by itself the main shortcoming of a paper that is, otherwise, well written and (at least from my point of view) substantially deprived of any shortcoming.

In fact, I've only two suggestions for improvements:

1) Authors should stress in the discussion whether this sampling strategy may eventually evolve or not in some sort of screening option, because of the rarity of this disorder, or conversely this study may only lead to a confirmatory test in very selected patients;

2) please note that something has gone wrong with figures 1 and 2. Even though the association of histogram bars with dhdh1 +/+ or -/- is quite obvious, the color scheme is not properly reported and should be refined.

Author Response

Thank you for the opportunity to review this interesting and innovative paper.

In this research study, Authors have addressed the option for new and innovative testing strategies based on the assessment of potential biomarkers for the spastic paraplegia type 28. The rarity of this disorder, and the difficulties experienced in obtaining valuable samples is appropriately addressed across the main text, and the very rare occurrence of this disease is by itself the main shortcoming of a paper that is, otherwise, well written and (at least from my point of view) substantially deprived of any shortcoming.

> Thank you for appreciating our paper! All changes from the first draft are highlighted in yellow. We responded to all comments as follows:

In fact, I've only two suggestions for improvements:

1) Authors should stress in the discussion whether this sampling strategy may eventually evolve or not in some sort of screening option, because of the rarity of this disorder, or conversely this study may only lead to a confirmatory test in very selected patients;

> I believe that the establishment of the blood biomarker for SPG28 will be useful not only as a blood test to determine HSP type but also as a platform for therapeutic drug development. I added a description of this at the end of the discussion (L391-397).

2) please note that something has gone wrong with figures 1 and 2. Even though the association of histogram bars with dhdh1 +/+ or -/- is quite obvious, the color scheme is not properly reported and should be refined.

> We apologize for any inconvenience caused. The fill seems to have disappeared when converting to PDF. I have corrected Figures 1 and 2 as noted. (Due to adding new figure, Figure1 and 2 have been changed to Figure 2 and 3 respectively.)

Reviewer 3 Report

LPA is a phospholipid that is abundant in plasma. LPA is also known to be a potential biomarker for ovarian cancer since LPA in plasma is shown to be specifically increased in ovarian cancer patients. Although LPA is the product of a reaction catalyzed by DDHD1, the total amount of LPA was increased in Ddhd1(-/-) mice. Since abnormal PI metabolism is likely to be the common pathological mechanism for these types of HSPs, peripheral PI levels are good candidates for biomarkers of other HSPs sharing the pathogenesis with SPG28. In my opinion, the method and whole manuscript are good and deserved publication.

In Figure 1 and 2, Ddhd1(+/+) and Ddhd1(-/-) did not differ and the author need to correct this.

Author Response

LPA is a phospholipid that is abundant in plasma. LPA is also known to be a potential biomarker for ovarian cancer since LPA in plasma is shown to be specifically increased in ovarian cancer patients. Although LPA is the product of a reaction catalyzed by DDHD1, the total amount of LPA was increased in Ddhd1(-/-) mice. Since abnormal PI metabolism is likely to be the common pathological mechanism for these types of HSPs, peripheral PI levels are good candidates for biomarkers of other HSPs sharing the pathogenesis with SPG28. In my opinion, the method and whole manuscript are good and deserved publication.

> Thank you for appreciating our paper! All changes from the first draft are highlighted in yellow. We responded to your comment as follows:

In Figure 1 and 2, Ddhd1(+/+) and Ddhd1(-/-) did not differ and the author need to correct this.

> Thank you for bringing up this important point. I have corrected Ddhdl (+/-) to Ddhd1 (+/+) in Table1, 2 (Due to adding new Table, Table1 and 2 have been changed to Table 2 and 3.)
